# Increased Pre-Boundary Lengthening Does Not Enhance Implicit Intonational Phrase Perception in European Portuguese: An EEG Study

**DOI:** 10.3390/brainsci13030441

**Published:** 2023-03-04

**Authors:** Ana Rita Batista, Dinis Catronas, Vasiliki Folia, Susana Silva

**Affiliations:** 1Center for Psychology at University of Porto, Faculty of Psychology and Educational Sciences, Psychology Department, University of Porto, Rua Alfredo Allen, s/n, 4200-135 Porto, Portugal; 2Lab of Cognitive Neuroscience, School of Psychology, Aristotle University of Thessaloniki, University Campus, 54124 Thessaloniki, Greece

**Keywords:** prosodic phrasing, prosodic boundaries, closure positive shift, boundary perception, pre-boundary lengthening, implicit boundary recognition task

## Abstract

Prosodic phrasing is the segmentation of utterances into prosodic words, phonological phrases (smaller units) and intonational phrases (larger units) based on acoustic cues—pauses, pitch changes and pre-boundary lengthening. The perception of prosodic boundaries is characterized by a positive event-related potential (ERP) component, temporally aligned with phrase boundaries—the Closure Positive Shift (CPS). The role of pre-boundary lengthening in boundary perception is still a matter of debate: while studies on phonological phrase boundaries indicate that all three cues contribute equally, approaches to intonational phrase boundaries highlight the pause as the most powerful cue. Moreover, all studies used explicit boundary recognition tasks, and it is unknown how pre-boundary lengthening works in implicit prosodic processing tasks, characteristic of real-life contexts. In this study, we examined the effects of pre-boundary lengthening (original, short, and long) on the EEG responses to intonational phrase boundaries (CPS effect) in European Portuguese, using an implicit task. Both original and short versions showed equivalent CPS effects, while the long set did not elicit the effect. This suggests that pre-boundary lengthening does not contribute to improved perception of boundaries in intonational phrases (longer units), possibly due to memory and attention-related constraints.

## 1. Introduction

Prosody refers to the melodic (pitch movements), timbral (voice quality) and rhythmic events (pauses, changes in the velocity of speech like pre-boundary lengthening) that coexist with speech sounds (vowels and consonants) in natural speech [1]. Prosodic processing has been long since acknowledged as an important component of speech perception [2]. Prosody may convey affective (i.e., expression of emotions) and/or linguistic information. Prosody-related linguistic information enables the segmentation of utterances into speech units with acoustically defined boundaries [3,4,5,6,7], which is commonly referred to as prosodic phrasing. Prosodic words, phonological phrases (relatable to clause components like noun or verb phrases) and intonational phrases (relatable to clauses) form a hierarchy of prosodically defined speech units [1,4,8]. The role of prosodic phrasing in speech processing is twofold: it allows the segmentation of larger speech units into smaller ones for chunking purposes, and it provides a context to solve syntactic or semantic ambiguities in speech, as in garden-path sentences [1,9,10,11,12]. Prosody is not the sole contributor to speech segmentation—syntax and discourse context also play a role—but it is a fundamental one [13].

The perception of prosodic units has been a topic of interest to linguistics, psycholinguistics and, more recently, to cognitive neuroscience. In 1999, Steinhauer and colleagues [12] found a neural correlate of phrase boundary perception in Event-Related Potential (ERP), opening a new strand of research on speech processing. In this first study [12], the authors found that sentences with two intonational phrase boundaries (B versions) elicited two positive deflections, while sentences with one boundary (A versions of the same sentences) elicited only one. Since these positive deflections were temporally aligned with intonational phrase boundaries, the ERP component was taken to reflect a phrase closure mechanism and was termed Closure Positive Shift (CPS). The discovery was replicated in other languages such as English (e.g., [14]), German (e.g., [15]) and Dutch (e.g., [16]). CPS was also observed in silent reading, i.e., without acoustic input [17]. Critically, Pannekamp et al. [18] showed that CPS is also elicited in the absence of syntactic, semantic, and lexical content in the sentences, a finding that directly links CPS with the prosodic aspects of phrase boundary processing.

The literature on the perception of Intonational Phrase (IPH) boundaries emphasizes three prosodic boundary cues: pause (silent interval after the last word in the phrase boundary), pitch change, and pre-boundary lengthening (extension of the final word or syllable in the phrase) [5,14,19]. Phrase boundaries containing these cues are more easily identified and considered more salient when compared to sentences lacking prosodic boundary markers [7]. However, the extent to which each acoustic cue or cue combinations contribute to IPH boundary perception remains unclear.

Available findings regarding pre-boundary lengthening cues suggest that these modulate phrase boundary perception, at least when coupled with boundary-related pitch information. Most studies have focused on phrasal units at the phonological phrase level (clause constituents), and all point to a relevant contribution from pre-boundary lengthening, equivalent to that of other cues (pitch, pause). Scott [20] manipulated pre-boundary lengthening in natural speech phonological phrases containing pitch information (e.g., (Kate) or Pat and Tony will come vs. (Kate or Pat) and Tony will come) and found that it modulates boundary recognition. Aasland and Baum [8], found that increasing pre-boundary lengthening in natural speech modulates behavioral recognition of phonological phrases (e.g., (Pink and black) and green vs. (Pink) and black and green) in neurotypical participants as efficiently as pauses and pitch markers. Holzgrefe-Lang et al. [19] used a combination of ERP and behavioral measures to investigate cue weighting in the recognition of phonological phrases (e.g., (Mona) or Lena and Lola vs. (Mona or Lena) and Lola). They manipulated pitch and pre-boundary lengthening independently and asked participants to perform an explicit boundary recognition task while the EEG was recorded. They found that a combination of pitch change and pre-boundary lengthening is necessary to elicit CPS. In contrast, at least one study using larger units—intonational phrases (e.g., (If you want to keep ahead,) it is very necessary to take time to do exercises)—found that the pause was a more powerful perceptual cue than both final lengthening and pitch cues, with the latter two cues showing up as perceptually equivalent [7].

Available findings suggest, thus, that the role of pre-boundary lengthening may differ across phonological phrases and intonational phrases, being weaker in the latter. This would be in line with findings of slightly different EEG responses to phonological phrases vs. intonational phrases [4], suggesting that these two units may not be equivalent when it comes to boundary recognition mechanisms. On the other hand, research on acoustic cues has consistently used explicit tasks (recognize boundary structure), but not implicit ones—as in the original CPS studies [12,18], where listeners were asked to perform a prosody-unrelated task while listening to sentences. Therefore, it is yet unknown if pre-boundary lengthening remains a relevant cue for prosodic phrasing in intonational phrase units and implicit (inattentive) prosodic processing, which is likely the most realistic context of speech perception, since listeners in natural settings tend to focus on lexico-syntactic information.

In the present study, we investigated the role of pre-boundary lengthening in the implicit detection of intonational phrase boundaries as measured by the CPS component. To that end, we used the original CPS paradigm [12,18], in which natural speech sentence pairs with one (version A) vs. two IPH boundaries (version B), reflecting clause-like constituents, that are presented to participants while they perform a lexical recognition task. The principle embedded in this paradigm is that responses to A vs. B versions should not differ in the first IPH boundary (IPH1, common to both), but only in the second IPH boundary (IPH2, available in B but not in A). From this viewpoint, a CPS effect means that the B–A difference is positive at the IPH2 boundary and larger than the B–A difference in the IPH1 boundary. As a first necessary step to proceed with our ultimate goal, we investigated whether the CPS effect was present in our original set of natural speech stimuli. Consequently, in order to determine the effect of pre-boundary lengthening (ultimate goal), we manipulated the original set of sentence pairs twice: first, by reducing the amount of pre-boundary lengthening (short set), secondly, by increasing it (long set). In both manipulations, we kept the pitch- and pause-related cue values of the original set. According to our hypothesis, if increased pre-boundary lengthening enhances boundary detection, we should see increased CPS effects in long compared to original, and in original compared to short. If the opposite holds true (longer prosodic units like IPHs and/or implicit tasks diminish the impact of pre-boundary lengthening) other patterns may be expected.

## 2. Materials and Methods

### 2.1. Participants

According to a priori power analysis, we would need at least 28 participants to capture a medium effect size with 80% power and a critical alpha of 0.05. Fifty-four native speakers of European Portuguese enrolled in this study. Thirteen were excluded due to excessive EEG artifacts (more than 30% of contaminated trials, n = 6) or outlier voltage values (n = 7). The final sample consisted of 41 participants (31 female, 10 male), aged 18-45 (M = 21.8; SD = 5.88), with a mean of 13.7 years of formal education (SD = 2.27; range 11–20). None reported hearing problems or epilepsy. All participants gave informed consent according to the declaration of Helsinki. The project was approved by the ethics committee of the Faculty of Psychology and Educational Sciences of the University of Porto (Ref. 2022/01-10).

### 2.2. Stimulus Materials

Following the CPS paradigm, we created a set of 48 European Portuguese sentence pairs, which either had the potential to generate one phrase boundary (two clauses, A version) or two (three clauses, B version). These were syntactically simple declarative sentences composed of high-frequency words (frequency data taken from the Porlex database [21]), verbs, and other syntactic constituents such as “and” or “but”. The two sentences in each pair had similar lexical content, and they were matched for the number of words and syllables (for A versions, mean number of syllables = 25.0; SD = 2.4; for B versions, mean number of syllables = 25.3; SD = 2.1). In the A versions, sentences contained one potential phrase boundary at an early position in the sentence, and in the B version an additional one at a later position. These sentence pairs were read by a native Portuguese speaker (female) in a sound booth and digitally recorded at 24 bit a sampling rate of 48 kHz. All files were normalized to +70 dB rms. An example of each version (A vs. B) is provided below ((1); # denotes phrase boundaries and IPH1/2 the identity of the preceding IPH; the complete list of sentences is presented in Appendix A).

(1):

A: (O João comprou carne) #IPH1 (o Jorge e a Luísa trouxeram saladas e bebidas). João bought meat # Jorge and Luísa brought salads and drinks.

B: (O João comprou carne) #IPH1 (O Jorge trouxe saladas) #IPH2 (e a Luísa trouxe bebidas). João bought meat # Jorge brought salads # and Luísa brought drinks.

Since a one-to-one mapping of syntactic units onto prosodic ones is not mandatory [22], we made perceptual and acoustic validations of prosodic structure. Prior to running the experiment, the initial pool of 48 pairs (AB) of spoken sentences was rated for the clarity in number of IPHs by four independent annotators (judges), among whom there was a foreign listener (naive to the Portuguese language). Annotators were asked to state whether A versions had clearly two IPHs, and whether B versions had three by answering Yes, No or Not sure (Appendix B). In cases where more than one annotator answered No or Not sure to one or both versions of a sentence, the pair was rejected. The final selection of 30 AB pairs was made. The set was acoustically validated for differences between the IPH boundaries of A vs. B versions regarding pause length and pitch change (expected to be equivalent at IPH1 but not IPH2). Pauses at IPH2 (version B) had an average length of 361 ms (SD = 118 ms), while in version A they were undetectable at the corresponding locations. Pause length at the end of IPH1 was similar across the two versions (A: M = 364; SD = 110; B: 384 ms, SD = 124 ms). Pitch cues were measured by computing the change in fundamental frequency in the last 200 ms of the IPH. As expected, the analysis showed rising pitch trends for stimuli at IPH2 in B versions (M = 78.81 Hz) that were not seen in the A version at the same position (M = 1.24 Hz).

The critical validation concerned pre-boundary lengthening at the end of IPH1 (version A and B) and IPH2 (version B only, Figure 1). Pre-boundary lengthening was defined as a larger-than-one ratio between the last stressed syllable of the IPH (in which pre-boundary lengthening is expected to begin) and the first stressed syllable of the sentence. Note that, in line with Oyedeji et al. [23], we assume that boundary cues (pitch and lengthening) start to appear in the last stressed syllable of the IPH (Figure 1), at least in European Portuguese (EP). In EP, the last stressed syllable of IPHs is not always the last syllable of the word. Instead, stress tends to occur at the penultimate syllable (trochaic pattern, though other situations exist). Since EP has, at least, partly, a stress-based rhythm, post tonic syllables (i.e., the last syllable, or the two final syllables) are usually marked by vowel reduction (conversion to schwa, e.g., salada becomes salade) or deletion (elimination of vowel, e.g., carn for carne), except in highly specific statements such as greeting calls [24]. Therefore, applying length changes in the last syllable would make the word sound unnatural, as if only the consonant had been lengthened.

Example (2) indicates the position of the first syllable of the sentence (upper case) and that of the last stressed syllable in the IPH (underlined) 

(2):

A: (O JoÃO comprou carne) # (o Jorge e a Luísa trouxeram saladas e bebidas). João bought meat # Jorge and Luísa brought salads and drinks.

B: (O JoÃO comprou carne) # (O Jorge trouxe saladas) # (e a Luísa trouxe bebidas). João bought meat # Jorge brought salads # and Luísa brought drinks.

The original set was edited twice, generating two additional sets: short and long. In the short set, pre-boundary lengthening in both A and B versions was set to half (in most cases eliminating pre-boundary lengthening). In the long set, pre-boundary lengthening was doubled. As a result of the transformations made in IPH1, IPH2 boundaries (B versions) were time-shifted in short (earlier) and long sentences (later, Figure 2). These manipulations were made using the software Praat (https://www.fon.hum.uva.nl/praat/ Amsterdam, The Netherlands; Version 6.51.52, accessed on 10 October 2021), specifically by creating a duration tier wherein we marked the time limits of the last stressed syllable and multiplied (long) or divided (short) the original duration by a factor of 2. Stimuli were, then, resynthesized, generating additional audio files with the desired transformations.

### 2.3. Procedure

Participants were instructed to listen to the sentences while performing a vigilance task, which was unrelated to the experimental manipulation, i.e., they did not perform explicit judgments about prosodic phrase boundaries. Participants listened to several sentences heard from the speakers connected to the stimulation computer. At the end of each sentence, a word appeared on the screen. Participants were then asked to judge if each of the words was part of the previously heard sentence or not, pressing two different keys on the computer keyboard (YES or NO). After receiving the instructions, participants performed practice trials and possible doubts were clarified. Counterbalancing across participants was performed by switching the label of the key numbers on the computer keyboard, creating two versions of the task, 1 (YES/NO) and 2 (NO/YES). For each of these two versions, we created two variants by pseudo randomizing the order of trials. The goal was to avoid consecutive presentations of the A and B versions of the same nuclear sentence or two length-related conditions of the same sentence.

Each trial was structured as follows: a fixation cross signaled the onset of the auditory presentation of the sentence; after the offset of the sentence, a blank screen was presented for 200 ms; the probe word appeared on the screen until a response was provided; and the response was followed by an interstimulus interval of 2000 ms, during which the screen was blank.

The experiment was run in an acoustically shielded room and lasted around 40 min, head preparation included.

### 2.4. EEG Recording and Preprocessing

Participants were seated in a comfortable chair in front of the stimulation computer. We then placed on their scalp an electrode cap with 64 active channels positioned according to the 10-20 system (FP1, FPz, FP2, AF7, AF3, AFz, AF4, AF8, F7, F5, F3, F1, Fz, F2, F4, F6, F8, FT7, FC5, FC3, FC1, FCz, FC2, FC4, FC6, FT8, T7, C5, C3, C1, Cz, C2, C4, C6, T8, TP7, CP5, CP3, CP1, CPz, CP2, CP4, CP6, TP8, P9, P7, P5, P3, P1, Pz, P2, P4, P6, P8, P10, PO7, PO3, POz, PO4, PO8, O1, Oz, O2, Iz). Two external electrodes placed at the mastoids were added to allow re-referencing during preprocessing. An additional electrode was placed below the left eye to record vertical eye movements (VEOG).

EEG data was collected using a Biosemi ActiveTwo system (https://www.biosemi.com/ Amsterdam, The Netherlands; Accessed on 1 July 2021) with 512 Hz sampling rate. Before the experiment started, signal quality was checked and kept under the system-recommended thresholds. Participants were asked to move as little as possible and try to blink only between trials.

We preprocessed EEG data with the Fieldtrip toolbox [25] for MATLAB (https://www.mathworks.com/ Massachusetts, United States of America; Accessed on 15 August 2022). After trial definition based on sentence-onset triggers, trials with vertical and horizontal eye movement artifacts were marked based on visual analysis. Trials with other types of artifacts detectable by variance inspection were also marked, as well as defective channels. Contaminated trials were rejected, and bad channels were interpolated using nearest neighbor averaging. Clean trials were baseline corrected (200 ms pre-trigger), detrended, re-referenced to the mastoid electrodes, and band-pass filtered between 0.01 and 30 Hz. Finally, trials of each condition were averaged per subject, and then grand averaged.

Triggers were placed at the onset of each sentence, in line with previous approaches to CPS [14]. However, since we noted early divergences between A and B versions in this scenario, we adopted new, IPH-related baselines, by applying baseline correction to each trial at the 200 ms period preceding the two relevant events (see Results: the minimum boundary onset time for IPH1 (common to short, original and long), and the same for IPH2 (different time points across length conditions). We then extracted the time window between 150 ms and 650 ms post-boundary onset for both IPH1 and IPH2 and ran the statistical analysis. The IPH2 trigger point was based on B versions, where the boundary was present (Figure 2), and it was applied to the A versions. Thus, at this point, we were comparing presence (B) vs. absence (A) of a boundary.

### 2.5. Statistical Analysis

Time-averaged voltage values per subject and region of interest (nine regions: anterior, central, posterior x left, mid right) were extracted for the time windows between 150 and 650 ms post boundary onset (post onset of last stressed syllable).

First, we analyzed the CPS effect per length and topography, considering the B–A difference at IPH2 (expected to be higher than the one at IPH1 in case of significant effect) minus the B–A difference at IPH1 as dependent variable. The expected CPS effect (the previous value, expressing the interaction IPH x length) would, thus, consist of a positive value. Once observing length effects, we compared the length conditions two at a time to locate significant length-related differences. Finally, we moved into the analysis of IPH (1 vs. 2) x version (A x B) interactions in each length condition (short vs. original vs. long) to verify whether and where values were significant.

In all analyses, the critical alpha value was set to 0.05. Greenhouse Geiser corrections were made for sphericity violations. Complementary Bayesian analyses were run in case of marginal results. We calculated Bayes Factors (BF) with default priors to further investigate the alternative hypothesis over the null one (BF_10_), using the JASP software (https://jasp-stats.org/ Amsterdam, The Netherlands; Version 0.16.0, Accessed on 15 September 2022) [26]. Unlike traditional null-hypothesis-significance-testing, which relies on dichotomous information (significant vs. non-significant results), Bayes factors quantify the relative predictive performance of two alternative hypotheses (alternative vs. null, or null vs. alternative), measuring the strength of evidence in favor of one over the other [27,28]. Bayes factors are particularly relevant to strengthen claims of null effects and clarify marginal results, and this was how we have mostly used them in the present study. Following the heuristics provided in van Doorn et al. [28], we considered BFs between 1 and 3, 3 and 10, 10 and 30 and above 30 as weak, moderate, strong and very strong evidence in favor of the alternative hypothesis. While BFs above 1 support the alternative hypothesis, BFs below 1 indicate evidence in favor of the null hypothesis, and evidence here becomes stronger as values decrease: BFs between 1 and 0.33 provided weak evidence, between 0.33 and 0.10 moderate, between 0.10 and 0.03 strong, and below 0.03 very strong.

## 3. Results

### 3.1. CPS Effect

The repeated measures ANOVA with length, caudality and laterality as factors, and the CPS effect ((B–A at IPH2)—(B–A at IPH1)) as dependent variable (Figure 3) showed a significant main effect of length, *F*(1,40) = 3.40, *p* < 0.038, *η_p_*^2^ = 0.078, without further interactions with topographical factors. Positive values for short and original indicated the expected effect (B–A at IPH2 greater than B–A at IPH1), while the negative values for long suggest that the effect is, at least, absent.

### 3.2. Pairwise Comparisons across Length Conditions

Comparisons between short and original showed no significant differences *F*(1,40) = 0.041, *p* = 0.84, *η_p_*^2^ = 0.001. Long differed significantly from original *F*(1,40) = 4.66, *p* = 0.037, *η_p_*^2^ = 0.10 and marginally from short *F*(1,40) = 4.66, *p* = 0.052, *η_p_*^2^ = 0.091, both comparisons showing medium effect sizes. Bayes factors revealed strong evidence of differences between short and long (BF_10_ > 30) and no evidence for the comparison short–original (BF_10_ = 0.098), suggesting that both short and original elicit relevant differences from long.

### 3.3. Interaction IPH x Version Per Length

For the original stimulus set, the interaction was significant, *F*(1,40) = 5.87, *p* = 0.020, *η_p_*^2^ = 0.13, and we found the expected CPS effect: B showed higher voltages than A at IPH2, while the reverse happened at IPH1 (Figure 4).

For the short stimulus set, we found a similar scenario, though the interaction was marginal, *F*(1,40) = 3.65, *p* = 0.064, *η_p_*^2^ = 0.083. BFs, however, provided strong evidence for interaction (BF_10_ > 30).

For the long stimulus set, a different pattern emerged, with the interaction IPH x version losing significance, *F*(1,40) = 1.59, *p* = 0.22, *η_p_*^2^ = 0.038. Note that, despite the lack of significance, the expected direction of the CPS effect was even reversed, with A showing higher voltages than B at IPH2, while the reverse happened at IPH1 (Figure 4).

In summary, while longer-than-original pre-boundary length values appear to attenuate the CPS effect, shorter-than-original values do not seem to make a difference.

## 4. Discussion

In the current study, we wanted to determine whether increased pre-boundary lengthening leads to enhanced implicit intonational phrase boundary detection as measured with the CPS ERP component in European Portuguese (EP). To that end, we manipulated a set of natural speech sentence pairs both by reducing pre-boundary lengthening (short set) and enlarging it (long set). We found that pre-boundary lengthening seems to affect phrase boundary processing, but not in the expected way: while both short and original elicited similar CPS responses, responses to long did not show the CPS effect, differing from both short and original. Therefore, variations in pre-boundary lengthening did not have the expected effect on phrase boundary perception, and this may be accounted for by several reasons.

One reason (1) could be that, since other prosodic-boundary cues were available (at least pitch and pauses), listeners simply ignored the variations in lengthening when processing boundaries. This would explain the lack of difference between short and original. To explain the difference between original (CPS effect) and long (no CPS effect), we would have to hypothesize an additional mechanism wherein the enlarged lengthening that was applied to long versions was excessively unnatural, and these sounded like speech aberrations.

Besides the fact that listeners had other cues, why would listeners ignore pre-boundary lengthening in particular? Based on the literature (see introduction), our hypothesis was that dealing with large units (IPHs instead of phonological phrases, as in previous research) and/or deviating listeners’ attention to non-prosodic information (implicit instead of explicit task) could diminish the weight of pre-boundary lengthening in IPH boundary perception when compared to phonological phrases (increased weight), in line with [7] vs. [8,19,20]. Why would these differences matter? Concerning the use of IPHs (clause-like) instead of phonological phrases (clause-component-like), we may hypothesize that longer units (IPHs) make it harder to maintain a reference syllable length in memory for comparison with the last stressed syllable, where pre-boundary lengthening takes place: if we admit that the first syllable is indeed a reference, then IPHs would require a much larger time window for memory maintenance than simple phonological phrases. As a result, listeners would focus more on short-time-window cues (pitch change) or even absolute cues (pause) for boundary processing. The reason for implicit tasks having a hypothetical negative effect on the use of pre-boundary lengthening may be related to the previous reasoning: faced with larger and more complex amounts of information to process (IPHs), listeners would be more available for short-term or absolute (less memory-demanding) cues in an implicit task than in an explicit one—where they were prompted to focus on prosodic patterns and, thus, have more chances to rely on all available cues. One way to test this possibility would be to carry out the experiment with synthetic speech, such that all boundary cues except lengthening were removed.

Regarding the failure in obtaining a CPS effect in long sentences (unlike what happened in short and original), it may have occurred because artificial lengthening, made without any pitch-related compensation, made the pitch contour unnatural, and this caused the atypical response we saw. A way to test this could be comparing long sets with vs. without pitch corrections for length. Moreover, even though we followed the procedures described in a previous study on EP, we agree that the possibility of extreme manipulations cannot be ruled out. For example, the ratios obtained for lengthening may have been bloated due to the frequent sentence-initial speed-up, and thus the manipulations based on those ratios were too extreme. A counter argument for this possibility is that, if manipulations were too extreme, they would affect the difference in short-original too (besides original-long), in the sense that the difference would be either shocking or noticeable. For instance, with a lengthening of 2, the difference between original and short would be 2–1, 1 point of difference for lengthening ratios; if lengthening was 1.5 for original, it would be 0.75 for short −0.75 difference). We saw no differences in the ERPs for short vs. original sets, suggesting that listeners did not feel unnatural length reduction in short versions and, furthermore, they did not discriminate between short from original. Future studies should address this question and consider other ways of calculating the degree of lengthening, for example, by using average syllable durations as a reference point instead of IP-initial syllables.

Another reason (2) for the lack of differences between short and original sets may be that listeners—instead of ignoring the length cue—tolerated the difference between short and long, something that makes sense in light of sociolinguistic explanations. In European Portuguese, clear differences are found between north and south dialects concerning pre-boundary lengthening, with northern variants showing increased values [29]. Since Portugal’s capital (and most urban) city is in the center-south, upper-class dialects in the north tend to adapt to south features, including shorter pre-boundary lengthening. On the other hand, our experiment was run in the north, where most participants were, thus, exposed to both north (due to location) and south (due to dialect adaptation) variants. This may have placed short and original sentences at similar levels of familiarity, thus preventing listeners from perceiving the short sentences as unnatural. One way of testing this hypothesis would be running the experiment with southern participants, listening to our northern speaker. In case the short stimuli showed advantage over the original (northern) one, this would indicate that familiarity plays a role in how pre-boundary lengthening is used in phrasing.

Besides the suggestions for future studies presented above, other ideas arise from the limitations of this study. Perhaps the biggest limitation is that we did not compare phonological phrases with intonational phrases, nor implicit with explicit tasks. This precludes us proposing more solid interpretations of our findings and make such comparisons a priority for the near future. As also mentioned in the methods, we saw early divergences in the responses to A and B versions, but we should only see these divergences at the boundary of IPH2. The most likely cause for this is the fact that we used natural speech in both versions and, hence, it was difficult to achieve perfect acoustic equivalence between the two versions, especially when speech units were relatively long. Perhaps future studies can apply post-recording prosodic manipulations to make versions A and B identical up to the onset of the IPH2 boundary. Moreover, looking at the ERP waveforms, one may question whether the time windows used for analysis were too short. It is indeed possible that CPS responses were prolonged beyond this time window of interest. However, the waveforms also clearly indicate that the pattern found in the time window analysis remains till the end of the sentence. Finally, regarding our motivated choice to manipulate the last stressed syllable of the IPH, future studies could investigate what happens when the last syllable is manipulated instead.

Despite its limitations, this study is, to our knowledge, the first to determine the role of pre-boundary lengthening in the implicit recognition of intonational phrase boundaries, raising new questions to address in the future. Our findings suggest that prosodic units at different scales may recruit different types of acoustic cues when it comes to perceptual segmentation, namely that pre-boundary lengthening is not recruited in the perception of IPHs when other cues are available. Finally, it is also possible that sociolinguistic factors have a strong influence in the process of prosodic boundary recognition.

## Figures and Tables

**Figure 1 brainsci-13-00441-f001:**
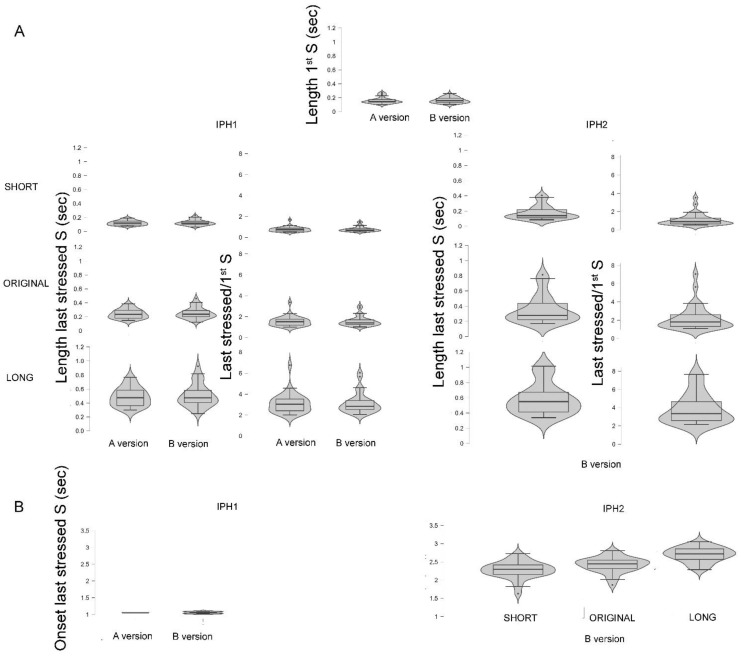
Properties of stimulus materials: (**A**) Length of first syllable (up), length of last stressed syllable and pre-final lengthening (length of last stressed/length of first) for IPH1 (down, left) and IPH 2 (down, right); (**B**) Onset time of last stressed syllable in IPH1 and IPH2. Note: IPH = intonational phrase; though IPH2 was present only in B versions, EEG analyses used B versions onset times in A for comparison between absent (**A**) and present boundary (**B**).

**Figure 2 brainsci-13-00441-f002:**
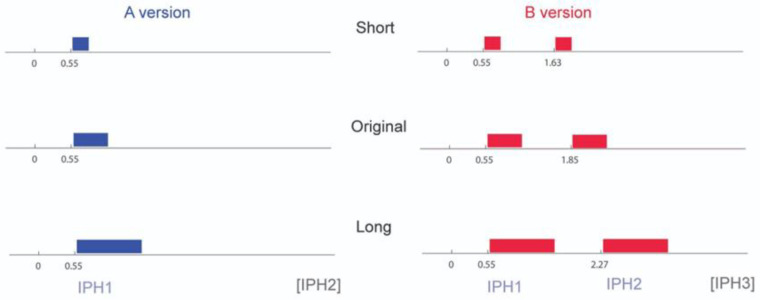
IPH boundaries (rectangles represent last stressed syllable of each IPH) at A and B versions, for short vs. original vs. long stimulus sets. For EEG analysis, IPH2 boundaries in A versions were copied from B versions.

**Figure 3 brainsci-13-00441-f003:**
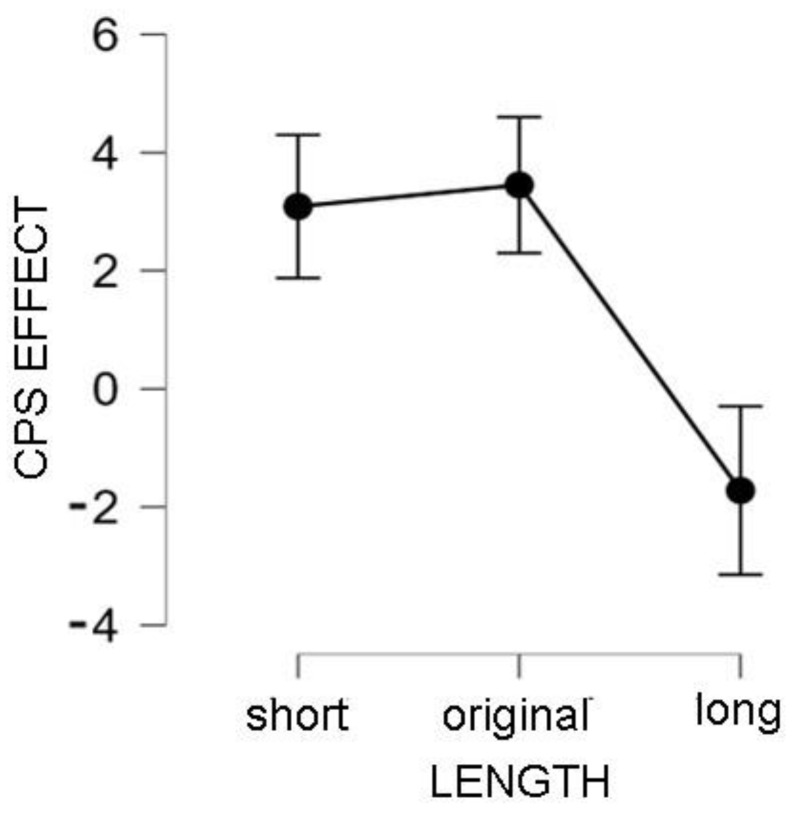
CPS effect across the three length (pre-boundary lengthening) conditions. Vertical bars represent 95% confidence intervals; The CPS effect refers to whole-scalp-averaged voltage values resulting from B–A at IPH 2 minus B–A at IPH 1 and is expected to be positive.

**Figure 4 brainsci-13-00441-f004:**
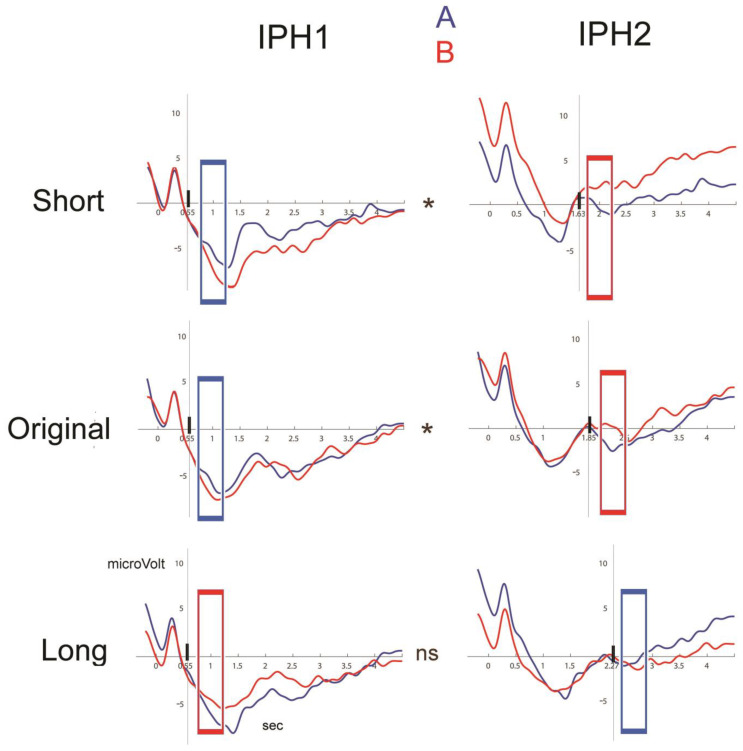
ERPs to IPH1 and IPH2 boundaries (onset of last stressed syllable in the IPH) across the three levels of pre-boundary lengthening. Baseline corrections were applied to the time points marking the onset of the last stressed syllable of both IPH1 and IPH2. The waveforms represent average voltage for the right-central region. Asterisks and ns (non-significant) refer to the significance of the IPH x version interaction (CPS effect). Rectangles indicate the 500 ms time window used for analysis: red rectangles indicate B > A and blue ones A > B.

## Data Availability

Data availability: stimulus materials (database and audio), as well as statistical analyses are available at osf link https://osf.io/gs3ep/?view_only=ae7fec4571b741ecbaca7f13a8e7df80.

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
