# Peer review of "Increased Pre-Boundary Lengthening Does Not Enhance Implicit Intonational Phrase Perception in European Portuguese: An EEG Study"

_brainsci, 2023, doi:10.3390/brainsci13030441_

Round 1
Reviewer 1 Report
Comments and Suggestions for Authors
In my view in this study some incorrect assumptions were made that severely influenced the construction of the stimuli and accordingly the results of the whole study.
It is maintained that pitch rise is one of the main prosodic IP boundary cues. Unfortunately, this is not necessarily the case, because the pitch rise as a boundary tone is found only in certain well-defined cases, as e.g. in questions, continuation rises, or lists.
I was surprised to see that F0 fall which is a frequent IP boundary marker, especially in statements was not mentioned at all. Relevant literature concerning this point was also not cited.
In my view, this incorrect assumption creates a serious problem for the experiment. More rising was found in version B of the data because most probably the participants treated the stimuli as a list, probably as an enumeration of the events, whereas this was not the case in Version A where the sentences were interpreted as separate events by the speaker.
Therefore, the results are not surprising, because the perception of a boundary does not only focus on a pre-boundary lengthening but includes other cues at the same time. The boundary was not perceived as such in the B experiment because it was meant as a continuation (a list), whereas in the short and normal cases, the informants heard separate IPs.
One possible way of checking the stimuli before running the experiment is to reach a high inter-rater agreement. The annotators (at least two) should listen to the stimuli and judge how strong the boundary is, see e.g. the usual annotation procedure in the TOBI system.
I strongly recommend inviting a prosodist to construct an appropriate data set to test the hypothesis put forward in the paper. If only a pre-boundary lengthening is of interest, then the stimuli should be the same differing only in the pre-boundary lengthening but not in the length of the sentences or the presence of other cues of different magnitude. The other parameters should be kept constant.
In summary, to test the current hypothesis a different stimulus set is needed. In order not to waste the recorded data, different hypotheses are required.
Author Response
#Rev1
In my view in this study some incorrect assumptions were made that severely influenced the construction of the stimuli and accordingly the results of the whole study.
R: Thank you for the valuable comments, suggestions and for raising these issues. We failed to communicate with clarity the principles guiding our stimulus construction, research paradigm, and their relation to our goal. We have now made changes according to the Reviewer’s comments in the introduction and discussion to give a clear overview in order to avoid further misunderstandings. A point-by-point detailed response is provided below.
1-It is maintained that pitch rise is one of the main prosodic IP boundary cues. Unfortunately, this is not necessarily the case, because the pitch rise as a boundary tone is found only in certain well-defined cases, as e.g. in questions, continuation rises, or lists.
R: Even though the direction of pitch changes at phrase boundaries is an important issue, it has little or no impact on our results: we did not manipulate pitch (nor pauses), and all cues related to these parameters were part of the natural speech samples we collected from the speaker, who uttered both A and B sentences in the original version. The IPHs were acoustically validated for pitch change, pauses and pre-boundary lengthening, as now clarified in the methods (lns 152-172). They were also perceptually validated. What we manipulated was IPH-final lengthening for both A and B versions, creating short and long sets for A+B versions. Therefore, whatever pitch cues we had in natural speech, they reflected the speaker’s intention to express one (A) or 2 (B) IPH boundaries, as induced by syntactic structure. This intention was acoustically and perceptually validated.
Of course, accuracy when referring to pitch changes is important. Thus, we have now corrected the term from “pitch rise” to “pitch change” (i.e. a pitch rise or pitch fall indicating the presence of a boundary tone and/or a pitch reset). wherever appropriate, as usually referred to in most of the literature We have evidence that non-final IPHs in European Portuguese do rely mostly on pitch rise, specifically on “continuation rise” [1], but, since this topic is irrelevant to our goal, we did not include this in the paper.
- Frota, S.; Vigário, M. Intonational phrasing in two varieties of European Portuguese. Tones and tunes 2007, 1, 265-291.
2-I was surprised to see that F0 fall which is a frequent IP boundary marker, especially in statements was not mentioned at all. Relevant literature concerning this point was also not cited.
R: Please see our response to the above comment.
3-In my view, this incorrect assumption creates a serious problem for the experiment. More rising was found in version B of the data because most probably the participants treated the stimuli as a list, probably as an enumeration of the events, whereas this was not the case in Version A where the sentences were interpreted as separate events by the speaker.
R: We did see increased pitch change (rise, in our case) in IPH2 of B versions, but this is exactly what we expected to see if there was an IPH2 boundary in version B, but not in version A. We have now clarified this at lns 159-167 (concerning the presence of pitch rise instead of fall at IPH boundaries, please see our response to comment 1).
The present paradigm has been used numerous times to demonstrate the perception of IPH boundaries, and our paper replicates the available findings, as we state in our introduction and discussion (lns 51-61,135, 333).
Regarding the potential difference between A and B (beyond the additional IPH at B), we have no reasons to believe that version A was interpreted as a set of separate events and version B as a list. We can understand the comment of the Reviewer only under the light of literature comparing structural phrase differences, for example a comparison between compounds (N1+N2+N3) and a list of events (N1, N2, N3) since indeed those two structures differ in prominence pattern and temporal structure [1]. However, in our case the sets did not differ in structure (N1+N2+N3) only in the amount of phrase boundaries: version A contained one phrase boundary at an early position (IPH1) in the sentence, and version B contained IPH1 and an additional phrase boundary at a later position (IPH2).
Having validated boundary perception (CPS effect) for the original set (2 in B, 1 in A), we moved on with comparisons between boundary perception at short, original and long - our ultimate research goal (see lns 111-116).
- Yuen, I.; Rattanasone, N.X.; Schmidt, E.; Macdonald, G.; Holt, R.; Demuth, K. Five-year-olds produce prosodic cues to distinguish compounds from lists in Australian English. Journal of Child Language 2021, 48, 110-128, doi:10.1017/S0305000920000227.
4-Therefore, the results are not surprising, because the perception of a boundary does not only focus on a pre-boundary lengthening but includes other cues at the same time. The boundary was not perceived as such in the B experiment because it was meant as a continuation (a list), whereas in the short and normal cases, the informants heard separate IPs.
R: We were not clear enough about the comparison between version A and B, and that between original, short and long sets. Our ultimate goal was to test whether pre-boundary lengthening (PBL) modulated phrase perception (via manipulating the original A+B version to create a short and a long pre-boundary duration set). To be able to achieve this, we needed to determine whether boundary perception occurred in the original set. In line with our expectations, we saw that listeners perceived a boundary at IPH2 from version B because there was a larger A-B difference in IPH2 (B has boundary, A has not) than in IPH1 (both A and B had boundaries). So, A-B comparisons and short-original-long comparisons (each of these including A+B) served different purposes. We clarified this at lines 102-121.
It is true that boundary perception includes more than one cue - potentially it includes pitch, pauses and PBL. We wanted to see whether PBL makes a difference when the others are present. We saw that it does not. This can mean either that (1) IPH-final lengthening was not used in boundary perception, or that it was used but participants accepted some variability in IPH-final lengthening (e.g., by accepting both short and long versions as representatives of south and north dialects, respectively). We added notes on this in the discussion, lns 337 onwards vs. 386 onwards.
5-One possible way of checking the stimuli before running the experiment is to reach a high inter-rater agreement. The annotators (at least two) should listen to the stimuli and judge how strong the boundary is, see e.g. the usual annotation procedure in the TOBI system.
R: We agree with the Reviewer’s comment, and that is why we had already established a check on the stimuli before running the experiment and confirmed a high inter-rater agreement by having more than one annotator (the stimuli were checked by at least four annotators). In addition, we also made sure to ask a foreign listener who is naive to the Portuguese language to judge how strong the boundary is and make a selection based on the feeling of 1 vs 2 boundaries from prosody.
We have now included this important information for reader clarity, lns 152-157. We do not think that the usual annotation procedure in the TOBI system will add extra in our work, but we will save this recommendation for future research.
6-I strongly recommend inviting a prosodist to construct an appropriate data set to test the hypothesis put forward in the paper. If only a pre-boundary lengthening is of interest, then the stimuli should be the same differing only in the pre-boundary lengthening but not in the length of the sentences or the presence of other cues of different magnitude. The other parameters should be kept constant.
R: We agree with the Reviewer on the way the stimuli have to be created, since it is exactly what we did. The only parameter that differed across A and B versions was the amount of phrase boundaries (1 or 2), The length (number of phonetic syllables) of A vs. B was equivalent. The point of having A and B was testing whether listeners perceived IPH boundaries, and we do have acoustic and EEG evidence that we succeeded in contrasting 1 boundary with 2 (A vs B). The critical manipulation corresponds exactly to what the Reviewer suggested: both the original sets of version A and B were edited for creating a long (doubling pre-boundary lengthening) and a short version (having pre-boundary lengthening) while keeping everything else constant. Thus, the length here does not refer to the length of the sentences but to the intonational lengthening. The other parameters were kept constant. These length manipulations of course lead to a lengthening of the sentence in the sense that some (intonational) cues are added (and not words), but our measure of interest (CPS, (B-A at IHP2) - (B-A at IPH1) is not affected by this.
7-In summary, to test the current hypothesis a different stimulus set is needed. In order not to waste the recorded data, different hypotheses are required.
R: We hope that, after disentangling the A and B versions from the 3 length-related conditions (original, short and long sets) above, that the Reviewer will agree that there is no need for a different stimulus set. Our ultimate goal was to test whether pre boundary lengthening modulated phrase perception (via manipulating the original set in a short and a long pre-boundary duration set). To be able to do that, we first compared the A vs B sentences in the original version to test whether participants are able to perceive IPH boundaries as expected within the CPS paradigm.
We have now included the above clarifications in the intro and discussion for reader clarity (lns 102-121, 327-336).
Reviewer 2 Report
Comments and Suggestions for Authors
The paper presents the results of an experimental study with speakers of European Portuguese that attempts to illuminate the importance of final lengthening as a cue for IP boundaries. The study suffers from a very limited scope, apparent gaps in the knowledge of the prosodic literature, and, crucially, severe methodological flaws.
My main point of criticism is that it makes little sense to manipulate syllable durations when an abundance of other cues (prosodic as well as syntactic and semantic) exist for listeners to chunk speech. An experiment built around durational manipulations is therefore not suited to assess whether changes in duration improve boundary detection. Prosodic cues that were present in the experiment (apart from lengthening) were pauses, F0 rise (in the B series), and possibly other cues not mentioned by the authors.
The phonological literature discusses at least two other cues for IP boundaries, which are not mentioned in the paper: an increased unit-initial speaking rate (also known as "initial rush"), and a change in voice quality. Unfortunately, the reader is not given access to the audio files, so I can only guess that most, if not all, of these cues were present in the stimuli, meaning listeners did not have to rely on durational cues at all (and were probably simply startled by the overlong words in the L condition).
This leads to another major confound: As the degree of lengthening was determined by dividing the duration of the initial syllable by the duration of the last stressed syllable, the ratios obtained for lengthening were likely bloated due to the effect of the initial speed-up, and the manipulations based on those ratios too extreme. It would have been safer to simply use average syllable durations as a reference point instead of IP-initial syllables.
Another problematic aspect is the fact that the last stressed syllable was used to calculate pre-boundary lengthening. Judging from the appendix, the position of the stressed syllable was not always at the same distance from the IP boundary. For that reason, it would have been more consistent to always manipulate the syllable directly preceding the IP end. There is ample cross-linguistic evidence that final lengthening is found on the final syllable regardless of the position of the stressed syllable, and stressed syllables may not be lengthened at all if they occur too far from the end of an IP.
The description of the methods leaves much to be desired. I am especially worried by the lack of information on how the stimuli were created. The authors write that "[t]he original set was edited twice, generating two additional sets: short and long", and that they used Praat, but how exactly did they proceed? This is absolutely crucial information that needs to be explained in detail. As mentioned before, the audio files should be uploaded somewhere to give the reader the possibility to assess the quality and naturalness of the stimuli themselves.
The authors seem to assume a one-to-one mapping between syntax and prosody. While this may be correct for the simplistic sentences used in the study, predicting the prosodic shape of syntactic constituents in actual spontaneous speech is far from trivial, and numerous counterexamples exist whereby prosodic boundaries occur within syntactic phrases or phrases are juxtaposed without any prosodic boundary marking.
I would also encourage the authors to formulate their results more carefully. A small study with speakers of one language does in no way warrant a general statement such as "This suggests that pre-boundary lengthening does not contribute to improved perception of boundaries in intonational phrases" (from the abstract). For such a statement to be verified, hundreds of languages would need to be investigated rigorously. Please abstain from advertising results from a single language study as cross-linguistic valid truths.
Two minor points:
a) The first sentence in the introduction (l.32) is bizarre. The readers is told that humans are the only species that communicate using spoken language. Why is this relevant to the present study on final lengthening? And, by the way, final lengthening is not exclusive to humans and has been observed in animals, e.g. in bird song (Tierney et al. 2011).
b) "Note that, in line with Oyedeji et al. [26], we assume that the onset of IPH boundaries corresponds to the last stressed syllable of the IPH" (l.162) - I don't think this is how the term "boundary" is commonly defined. Boundaries are not domains, they separate and delineate domains.
Finally, some suggestions for further reading on final lengthening:
- Krivokapić (2007) on the effect of phrasal length complexity on pause duration
- Himmelmann et al. (2018) on the universality of IPs and phonetic cues associated with them
- Ladd (2008) on basic notions of intonational phonology, and the difficulties in defining prosodic phrases (pp.288-290)
- Paschen et al. (2022) on cross-linguistic patterns of final syllable lengthening
- Franz et al. (2022) for a discussion of the trade-off between pauses and final lengthening
-------------------------
Franz, Isabelle, Christine A. Knoop, Gerrit Kentner, Sascha Rothbart, Vanessa Kegel, Julia Vasilieva, Sanja Methner, Mathias Scharinger, Winfried Menninghaus. 2022. Prosodic Phrasing and Syllable Prominence in Spoken Prose. A Validated Coding Manual. Preprint: https://osf.io/h4sd5
Himmelmann, Nikolaus P., et al. 2018. On the universality of intonational phrases: A cross-linguistic interrater study. Phonology 35(2), 207-245.
Krivokapić, Jelena. 2007. Prosodic planning: Effects of phrasal length and complexity on pause duration. Journal of Phonetics 35(2), 162-179, 10.1016/j.wocn.2006.04.001.
Paschen, Ludger, Susanne Fuchs, Frank Seifart. 2022. Final Lengthening and vowel length in 25 languages. Journal of Phonetics 94, 101179.
Tierney, A.T., F.A. Russo, A.D. Patel. 2011. The motor origins of human and avian song structure. Proceedings of the National Academy of Sciences 108, 15510-15515
Author Response
#Rev2
The paper presents the results of an experimental study with speakers of European Portuguese that attempts to illuminate the importance of final lengthening as a cue for IP boundaries. The study suffers from a very limited scope, apparent gaps in the knowledge of the prosodic literature, and, crucially, severe methodological flaws.
R: Thank you for your valuable comments, which helped us improving the clarity of our manuscript and provide a more comprehensive discussion of the results.
1-My main point of criticism is that it makes little sense to manipulate syllable durations when an abundance of other cues (prosodic as well as syntactic and semantic) exist for listeners to chunk speech. An experiment built around durational manipulations is therefore not suited to assess whether changes in duration improve boundary detection. Prosodic cues that were present in the experiment (apart from lengthening) were pauses, F0 rise (in the B series), and possibly other cues not mentioned by the authors.
R: We agree with the Reviewer that listeners had other cues, and we mention these in the paper (pitcg change and pauses). Nevertheless, we followed the standard experimental reasoning, and we modified durational cues while keeping the others constant across length levels (short, original and long IPH-final lengthening). Our goal was to see whether durational cues contributed to IPH boundary perception in implicit prosodic processing (an open question), and we had no reason to believe that listeners would discard durational cues when other cues are present. This may have happened, though. We have now explored this possibility in the discussion (lns 337-339). The only alternative to address the role of durational cues (without leaving the others in the stimuli) would be using synthetic speech, or heavy audio processing to cancel pitch movements and pauses (and possibly others, as you mention). In our opinion, this was not the best choice to investigate natural speech, as we have now clarified.
We have now included the above reasoning with the relevant literature in the discussion (lns. 363-365) for reader clarity.
2-The phonological literature discusses at least two other cues for IP boundaries, which are not mentioned in the paper: an increased unit-initial speaking rate (also known as "initial rush"), and a change in voice quality. Unfortunately, the reader is not given access to the audio files, so I can only guess that most, if not all, of these cues were present in the stimuli, meaning listeners did not have to rely on durational cues at all (and were probably simply startled by the overlong words in the L condition).
R: We have now made generic references to voice quality (introduction) and initial rush (discussion). Unfortunately, we did not have the time to reanalyse the stimuli for these two additional cues, but we will save this recommendation for future analyses.
The Reviewer has been given access to the audio via the osf link (lns 439-441). As stated above, we agree with the Reviewer that it is possible that the listener did not rely on duration and relied only on pitch and/or pause (see lns 363 onwards).
3-This leads to another major confound: As the degree of lengthening was determined by dividing the duration of the initial syllable by the duration of the last stressed syllable, the ratios obtained for lengthening were likely bloated due to the effect of the initial speed-up, and the manipulations based on those ratios too extreme. It would have been safer to simply use average syllable durations as a reference point instead of IP-initial syllables.
R: We agree that the possibility of extreme manipulations cannot be ruled out and we added this important comment to the discussion (lns 366-386) to be addressed in future studies.
On the other hand there is a possibility if the manipulations were too extreme, that they would affect the difference short-original too (besides original-long), in the sense that the difference would be quite obvious. For instance, with a lengthening of 2, the difference between original and short would be 2-1, 1 point of difference for lengthening ratios; if lengthening was 1.5 for original, it would be 0.75 for short -0.75 difference). However, we saw no differences in boundary phrase perception for short vs. long. This may be due to the fact that length was not used, but it could also be the case that manipulations were not extreme, at least in this case.
4-Another problematic aspect is the fact that the last stressed syllable was used to calculate pre-boundary lengthening. Judging from the appendix, the position of the stressed syllable was not always at the same distance from the IP boundary. For that reason, it would have been more consistent to always manipulate the syllable directly preceding the IP end. There is ample cross-linguistic evidence that final lengthening is found on the final syllable regardless of the position of the stressed syllable, and stressed syllables may not be lengthened at all if they occur too far from the end of an IP.
R: It is true that the last stressed syllable of our IPHs is not always the last syllable of the word. In European Portuguese (EP), stress tends to occur at the penultimate syllable (though other situations exist). Since EP has, at least partly, a stress-based rhythm, post tonic syllables (i.e. the last syllable, ot the two final syllables) are usually marked by vowel reduction (conversion to schwa, e.g., salada becomes salade) or deletion (elimination of vowel, e.g., carn for carne), except in highly specific statements such as greeting calls. Therefore, applying length changes in the last syllable would create an unnatural sonority, associated to the lengthening of a consonant only.
We added this justification to the manuscript (lns 415-417).
5-The description of the methods leaves much to be desired. I am especially worried by the lack of information on how the stimuli were created. The authors write that "[t]he original set was edited twice, generating two additional sets: short and long", and that they used Praat, but how exactly did they proceed? This is absolutely crucial information that needs to be explained in detail. As mentioned before, the audio files should be uploaded somewhere to give the reader the possibility to assess the quality and naturalness of the stimuli themselves.
R: We have now specified the kind of manipulations we did (lns 199-202), which were based on marking target points (onset and offset of last stressed syllable) in a duration tier (menu “manipulation”), doubling or halving the syllable duration, and resynthesizing the audio. As stated above, the stimuli have been provided in the osf repository.
6-The authors seem to assume a one-to-one mapping between syntax and prosody. While this may be correct for the simplistic sentences used in the study, predicting the prosodic shape of syntactic constituents in actual spontaneous speech is far from trivial, and numerous counterexamples exist whereby prosodic boundaries occur within syntactic phrases or phrases are juxtaposed without any prosodic boundary marking.
R: Our intention was not to imply a one-to-one mapping between syntax and prosody, we thank the Reviewer for highlighting this important point. On the contrary, we sought to demonstrate that we had IPHs (prosodic units, acoustically and perceptually validated), independent from the clause structure of sentences. This is why we use the terms “clause-like” and so on. We have now made changes to clarify this (e.g., ln 152). We also made changes to the abstract to highlight the main point, which is the difference between smaller (phonological phrase) and larger units (intonational phrase). We hope that we have clarified this adequately.
7-I would also encourage the authors to formulate their results more carefully. A small study with speakers of one language does in no way warrant a general statement such as "This suggests that pre-boundary lengthening does not contribute to improved perception of boundaries in intonational phrases" (from the abstract). For such a statement to be verified, hundreds of languages would need to be investigated rigorously. Please abstain from advertising results from a single language study as cross-linguistic valid truths.
R: We have now clarified that conclusions of the study only apply to European Portuguese (e.g., ln 329). We have also stated this important Reviewer’s comment in our Title and abstract.
Two minor points:
a) The first sentence in the introduction (l.32) is bizarre. The readers is told that humans are the only species that communicate using spoken language. Why is this relevant to the present study on final lengthening? And, by the way, final lengthening is not exclusive to humans and has been observed in animals, e.g. in bird song (Tierney et al. 2011).
R: We agree with the Reviewer and have already removed that sentence.
b) "Note that, in line with Oyedeji et al. [26], we assume that the onset of IPH boundaries corresponds to the last stressed syllable of the IPH" (l.162) - I don't think this is how the term "boundary" is commonly defined. Boundaries are not domains, they separate and delineate domains.
R: The word boundary seems to be used with different meanings in the literature. It is true that a boundary is expected to be like a line, but references to boundary onset and offset are also available, namely referring to the onset and offset of a silent pause.
What we meant with that expression was that IPH cues (of which lengthening is part) typically start at the last stressed syllable, at least in European Portuguese (see above our response to your comment on not using the last syllable). We have now reworded the sentence (ln 173).
Finally, some suggestions for further reading on final lengthening:
- Krivokapić (2007) on the effect of phrasal length complexity on pause duration
- Himmelmann et al. (2018) on the universality of IPs and phonetic cues associated with them
- Ladd (2008) on basic notions of intonational phonology, and the difficulties in defining prosodic phrases (pp.288-290)
- Paschen et al. (2022) on cross-linguistic patterns of final syllable lengthening
- Franz et al. (2022) for a discussion of the trade-off between pauses and final lengthening
R: We kindly thank the Reviewer. We have now incorporated (when appropriate) the relevant literature. However, our paper is based on EEG and CPS effect instead of behavioral linguistic findings, so not all references could be used.
Reviewer 3 Report
Comments and Suggestions for Authors
There’s nothing wrong with this manuscript. The manuscript reads quite well, and the experimental design and analyses are good enough. However, as implicitly stated in the manuscript, this research may not provide a novel finding that will contribute to the field. The authors attempted to explain why both short and original elicited similar CPS responses, but the similarity might have been attributed to a stimuli design or something like that. Although the authors tried to discuss the advantage of both original and short versions over long ones, I think the discussion is based on convincing evidence. The wildest possible speculations were made to describe the findings of the research. Regarding these issues, the authors need to state them in the introduction section and try to think about a possible direction for the current study. Again, the authors had better consider whether the similarity between the short and original is a kind of novel finding in this research.
Author Response
#Rev3
There’s nothing wrong with this manuscript. The manuscript reads quite well, and the experimental design and analyses are good enough. However, as implicitly stated in the manuscript, this research may not provide a novel finding that will contribute to the field.
R: Thank you for your valuable comments, which helped us improving the clarity of our goals vs. findings, and provide a more comprehensive discussion of the results.
1-However, as implicitly stated in the manuscript, this research may not provide a novel finding that will contribute to the field.
R: The purpose of the current study was to shed light on the particular contribution of pre-boundary lengthening to European Portuguese IPB processing, since its role in relation to the other cues (pause and pitch) is yet not well researched and debated in the current literature (see lns 90-121). We reviewed the literature and derived a hypothesis - pre-boundary lengthening could have little impact in longer units such as intonational phrases, and in implicit tasks. To our knowledge, this was the first study addressing this hypothesis, and we found evidence in support of it. We clarified this in the discussion (lns 344-365 and 418-423) and abstract.
2-The authors attempted to explain why both short and original elicited similar CPS responses, but the similarity might have been attributed to a stimuli design or something like that. Although the authors tried to discuss the advantage of both original and short versions over long ones, I think the discussion is based on convincing evidence. The wildest possible speculations were made to describe the findings of the research. Regarding these issues, the authors need to state them in the introduction section and try to think about a possible direction for the current study. Again, the authors had better consider whether the similarity between the short and original is a kind of novel finding in this research.
R: We have now improved the articulation between our hypothesis as stated in the introduction and our findings (please see our response to the above comment).
We have also added the possible interpretation that durational cues were simply not used (non-integrated in boundary perception) because there were other cues as an alternative to the one we had previously suggested (lengthening was integrated in boundary perception, but both short and original sets were found acceptable by listeners). The sociolinguistic interpretation as framed by the alternative idea - durational cues were used but, at least for short vs. original, listeners may have tolerated both as good-enough indicators of boundaries (lns 387-400).
Round 2
Reviewer 1 Report
Comments and Suggestions for Authors
Review:
Unfortunately, I cannot see significant improvements.
Since duration was an essential parameter on which the conclusions were based on, the measurements should be normalized. How does the paper account for the fact that speakers spoke with a different speech rate, a factor that most possibly affected the results. In addition that there were not examined from the point of view of F0 or intensity.
Other issues
1) The annotator agreement should be calculated and it should be reported what happened in the case of disagreement.
2) The sentence on page 14. L.180f makes a surprising assumption. Please consider that sonority has been never linked with duration. Where does this implication come from?
3) P.14, l.261: It remains unclear how IPH2 was calculated considering the fact that there were no results for IPH2 of A. The table cells are empty. Please clarify.
4) Table 1: It would be more transparent to present the table in forms of graphs.
5) Abstract: it is unknow -> it is unknown
Author Response
Response to Reviewer’s comments:
Unfortunately, I cannot see significant improvements.
Thank you for your comments. Below we provide a point-by-point response.
Since duration was an essential parameter on which the conclusions were based on, the measurements should be normalized. How does the paper account for the fact that speakers spoke with a different speech rate, a factor that most possibly affected the results. In addition that there were not examined from the point of view of F0 or intensity.
There was only one speaker included in the experiment, so there are no different speech rates from several speakers as we explain under section 2.2. Stimulus materials: “The original set of 30 A + 30 B sentences was read by a native Portuguese speaker (female) in a sound booth and digitally recorded at 24 bit a sampling rate of 48 kHz.” In section 2.3. Procedures, there is a plural of the word speaker, i.e., speakers but this refers to the computer speakers, the output hardware device that connects to a computer to generate sound. We have now corrected the term to “speakers connected to the stimulation computer”. So, F0 and intensity-related normalization is not necessary.
Additionally, we agree that F0 or intensity (pitch) changes at phrase boundaries are indeed a very important issue in the literature for those who choose to manipulate those parameters; however, we did not manipulate pitch (nor pauses), and these remained only as part of the natural speech samples collected from only one speaker, reflecting the speaker’s intention to express one (A) or 2 (B) IPH boundaries. Of course, these intentions might have failed, and this is why the number of IPHs (2 in A, 3 in B) was acoustically and perceptually validated. In conclusion, we did not treat F0 or intensity as factors in our EEG study. Only duration was manipulated, therefore, we kept them constant and controlled our stimuli for those possible confounders.
We acknowledge, though, that we did not report normalization values for intensity across the whole set. This could be relevant in case intensity was so low that participants could not understand the sentences. We did it now under section 2.2. Stimulus materials: “All files were normalized to +70 dB rms.”
Other issues
1) The annotator agreement should be calculated and it should be reported what happened in the case of disagreement.
We have now provided the ratings as an appendix (B), as well as the criteria for rejection: In case more than one annotator did not recognize the number of IPHs the sentence should have (or was unsure about it), the sentence was rejected. See section 2.2. Stimulus materials: “Prior to running the experiment, the initial pool of 48 pairs (AB) of spoken sentences was rated for the clarity in number of IPHs by four independent annotators (judges), among whom there was a foreign listener (naive to the Portuguese language). Annotators were asked to state whether A versions had clearly two IPHs, and whether B versions had three by answering Yes, No or Not sure (Appendix B). In cases where more than one annotator answered No or Not sure to one or both versions of a sentence, the pair was rejected.”
2) The sentence on page 14. L.180f makes a surprising assumption. Please consider that sonority has been never linked with duration. Where does this implication come from?
The term “sonority” was an unfortunate choice, since we were not referring to the hierarchy of loudness across sound classes (vowels, consonants etc.). We were just referring to the way words would sound (natural or unnatural). We have now corrected this. See section 2.2. Stimulus materials: “Therefore, applying length changes in the last syllable would make the word sound unnatural, as if only the consonant had been lengthened,”
3) P.14, l.261: It remains unclear how IPH2 was calculated considering the fact that there were no results for IPH2 of A. The table cells are empty. Please clarify.
The trigger point to analyze IPH2 at version A was borrowed from version B. Even though the boundary does not exist in version A (and therefore values were not reported in the table -now figure 1), the idea was to contrast the presence of a boundary (B) with the absence of it. We have now clarified this important point. See Figure 1, “Note: IPH = intonational phrase; though IPH2 was present only in B versions, EEG analyses used B versions onset times in A for comparison between absent (A) and present boundary (B)”; also Figure 2, Title: “IPH boundaries (rectangles represent last stressed syllable of each IPH) at A and B versions, for short vs. original vs. long stimulus sets. For EEG analysis, IPH2 boundaries in A versions were copied from B versions.”; also 2.4. EEG recording and preprocessing: “The IPH2 trigger point was based on B versions, where the boundary was present (Figure 2), and it was applied to the A versions. Thus, at this point, we were comparing presence (B) vs. absence (A) of a boundary.”
4) Table 1: It would be more transparent to present the table in forms of graphs.
We thank the Reviewer for this valuable suggestion. We have now transformed the table into graphs; it reads better now.
5) Abstract: it is unknow -> it is unknown
We thank the Reviewer for noticing, we have now corrected this.